# Transcriptional Regulation of Dental Epithelial Cell Fate

**DOI:** 10.3390/ijms21238952

**Published:** 2020-11-25

**Authors:** Keigo Yoshizaki, Satoshi Fukumoto, Daniel D. Bikle, Yuko Oda

**Affiliations:** 1Section of Orthodontics and Dentofacial Orthopedics, Division of Oral Health, Growth and Development, Kyushu University Faculty of Dental Science, Fukuoka 812-8582, Japan; yosizaki@dent.kyushu-u.ac.jp; 2Section of Pediatric Dentistry, Division of Oral Health, Growth and Development, Kyushu University Faculty of Dental Science, Fukuoka 812-8582, Japan; fukumoto@dent.tohoku.ac.jp; 3Division of Pediatric Dentistry, Department of Oral Health and Development Sciences, Tohoku University Graduate School of Dentistry, Sendai 980-8575, Japan; 4Departments of Medicine and Endocrinology, University of California San Francisco and Veterans Affairs Medical Center, San Francisco, CA 94158, USA; daniel.bikle@ucsf.edu

**Keywords:** transcription, transcription factor, stem cell, cell fate, lineage, dental epithelia, mediator, enamel

## Abstract

Dental enamel is hardest tissue in the body and is produced by dental epithelial cells residing in the tooth. Their cell fates are tightly controlled by transcriptional programs that are facilitated by fate determining transcription factors and chromatin regulators. Understanding the transcriptional program controlling dental cell fate is critical for our efforts to build and repair teeth. In this review, we describe the current understanding of these regulators essential for regeneration of dental epithelial stem cells and progeny, which are identified through transgenic mouse models. We first describe the development and morphogenesis of mouse dental epithelium in which different subpopulations of epithelia such as ameloblasts contribute to enamel formation. Then, we describe the function of critical factors in stem cells or progeny to drive enamel lineages. We also show that gene mutations of these factors are associated with dental anomalies in craniofacial diseases in humans. We also describe the function of the master regulators to govern dental lineages, in which the genetic removal of each factor switches dental cell fate to that generating hair. The distinct and related mechanisms responsible for the lineage plasticity are discussed. This knowledge will lead us to develop a potential tool for bioengineering new teeth.

## 1. Introduction

Tooth bioengineering is of great interest because dental decay and tooth loss constitute major public health issues, and tooth anomalies are commonly found in many craniofacial diseases. Compared to the success of dental pulp stem cells (SC) in regenerative medicine [1], it has been a great challenge to regenerate dental enamel, the hardest tissue of the body. Dental enamel is produced by dental epithelial SC and progeny residing in the tooth [2], and their cell fate is controlled by specific transcription program [3].

Transcription factors (TFs) are ultimate regulators to conduct cell specific transcription in every biological process [4]. They are expressed in specific cell types and regulate the expression pattern. They recognize specific DNA sequences called response element or TF binding site and activate or repress the gene expressions [4]. Some TFs called fate TFs serve as the major drivers specifying cell fate [4,5,6] by orchestrating the fate-specific transcriptional program.

Particular TFs possess the remarkable ability to reprogram one type of cell to another. The best known example is the combination of four TFs—Oct4(Pou5f1)/Sox2/Nanog/Klf4—that convert somatic cells to a pluripotent state [7]. Even one TF is sufficient to trans-differentiate somatic cells into another lineage. Myoblast determination protein (MyoD) converts fibroblasts to myoblasts [8]. Erythroid TF of GATA-binding protein 1 (Gata1) changes myeloblasts to erythrocyte precursors [9]. The CCAAT/enhancer-binding protein (Cebpα or β) converts B lymphocytes to macrophages [10].

In addition to TFs, the current model of transcriptional regulation includes a role for chromatin regulators to specify cell fate. They orchestrate gene transcription through controlling chromatin dynamics. For example, the chromatin remodeling complex, switching defective/sucrose non-fermenting factors (SWI/SNF) controls cell lineages through enhancer maintenance [11]. Highly conserved chromatin modifying complexes, such as the nucleosome remodeling and deacetylation (NuRD) complex, are also associated with lineage commitment during early development [12]. The special AT-rich sequence binding protein (SATB1) modulates the NuRD complex to regulate chromatin architecture and has the ability to modulate dental lineage [12]. The Mediator complex also controls cell lineage by facilitating gene transcription. The Mediator forms “super-enhancers” [13], which differ from typical enhancers in density and size. In super-enhancers, fate TFs are highly condensed to activate the transcription of cell identity genes [13]. For example, the Mediator complex maintains the cell fate of embryonic SCs (ESC) regulating four reprogramming TFs—Oct4(Pou5f1)/Sox2/Nanog/Klf4—within these super-enhancers. Reduced expression of the Mediator subunits induces ESC differentiation as a result of losing their pluripotent state following decreased expression of these 4 factors [13,14]. Mediator 1 (MED1) is one of the subunits of the multi-protein Mediator complex. Ablation of *Med1* in vivo results in embryonic lethality in mice, but conditional *Med1* null mice have been used to demonstrate its role in various cell lineages, including blood cells [15], T and B cells [16], and mammary epithelia [17,18]. Med1 controls epidermal lineages in skin, in which *Med1* ablation in *keratin 14* (*Krt14*) expressing epithelia enhances epidermal and sebaceous lineages while abolishing hair fate resulting in alopecia [19]. The same *Med1* null mice convert the dental lineage to skin epithelia in the tooth [20,21].

Understanding the transcriptional program controlling their cell fate is crucial to our efforts to build and repair teeth. Identification of master regulators controlling dental transcriptional regulatory networks is necessary for successful manipulation of pluripotent or adult SCs to regenerate dental enamel for tooth bioengineering. Therefore, the control of enamel cell fate in tooth development and regeneration is the main theme of this review. A number of factors have been identified that control the cell fate of enamel producing dental epithelium. In this review, we describe the current understanding of TFs and chromatin regulators controlling dental cell fate. We first describe the development and morphogenesis of mouse dental epithelia in (1) early development, (2) different dental lineages towards subpopulations such as enamel producing ameloblasts, and (3) adult SCs in incisor to regenerate dental epithelia postnatally. Then, we discuss the role of critical TFs or chromatin regulators by focusing on (1) SCs and their renewal, (2) commitment to different lineages, and (3) lineage plasticity. We also discuss the clinical significance of these factors through their gene mutations causing dental defects in craniofacial diseases in humans. Our main focus is on the epithelial TFs that have the re-programming potential to regenerate enamel. Several signaling pathways such as Wnt, FGF, TGFβ, and BMP are important but not mentioned in here as they have already been reviewed by others [22,23].

## 2. Development and Morphogenesis of Mouse Dental Epithelium

### 2.1. Initiation of Tooth Development

During embryonic development, tooth morphogenesis is initiated by thickening of dental epithelium to form a dental placode, followed by invagination into the mesenchyme in mice. Thereafter, tooth buds progress into the cap stage and primary enamel knots are formed in dental epithelium to lead to tooth cusps.

### 2.2. Dental Epithelial SC and Enamel Producing Epithelium

#### 2.2.1. Inner Enamel Epithelia (IEE) Lineage

IEE cells are important for tooth morphogenesis as they eventually differentiate to enamel-producing ameloblasts. The basement membrane (BM) that lies between the epithelium and mesenchyme is critical for IEE differentiation and tooth morphogenesis [24,25]. Adhesion molecules such as LAMA5 and LAMA2 are important for IEE and tooth morphogenesis [26,27]. Mutations in LAMA3 or LAMB3 cause amelogenesis imperfecta in humans [28,29]. Nephronectin (NPNT) is an ECM protein possessing 5 EGF-like repeat domains and a RGD sequence that promotes proliferation and differentiation of IEE. The NPNT localizing in the BM of the developing tooth reduces the number of SCs and increases cell proliferation at least partially through the EGF signaling pathway [30].

#### 2.2.2. Stratum Intermedium (SI) Lineage

Dental epithelial SC also differentiate into the SI lineage that is located adjacent to IEE cells and ameloblasts. SI cells support enamel mineralization by expressing alkaline phosphatase (ALPL) [20], which is essential for mineralization of the tooth and bone, as shown by hypo-mineralization in conditional *Alpl* null mice [31,32,33]. SI cells also express Notch1, which is central to their differentiation. Notch signaling is induced by Notch ligands Jag1 and Jag2, which are located in the adjacent IEE and ameloblasts [34], in which *Jag2*-deficient mice also show enamel hypoplasia [35]. A single-cell RNA-seq and lineage tracing suggests that SI cells possess high lineage plasticity as Notch1-expressing SI cells are converted to ameloblasts during injury induced regeneration [23,36].

#### 2.2.3. Outer Enamel Epithelia (OEE) and Stellate Reticulum (SR) Lineages

The OEE is fused with IEE at the crown cervical margin and forms Hertwig’s epithelial root sheath (HERS), which contributes to root formation in teeth [37,38]. A single-cell transcriptome study suggests that OEE cells control tooth size whereas SR cells regulate transport of nutrients in the incisor [39]. The unbiased clustering from single-cell analyses at 7 days old mouse incisors indicates that IEE and OEE, or SI and SR, are not much distinguishable by transcriptome. In addidtion, novel makers are identified in which ATF3 marks both OEE and SR cells. However, KRT15 labels only OEE [39]. However, independent single-cell transcriptome study at 8 weeks of the incisor demonstrates that dental epithelia are further divided into more than traditional category, in which OEE are clearly separated from upper IEE and IEE–OEE junctional region and further divided into two groups (OEE-1 and OEE-2). The SR is close to SI, and they are categorized into three groups: SI, inner SR/SI, and outer SR [36].

#### 2.2.4. Ameloblast Lineage

Ameloblasts are specialized epithelial cells responsible for the formation of the enamel, the hardest tissue in the human body. Ameloblast differentiation goes through a series of sequential morphological changes [40]. IEEs progress to presecretory ameloblasts. The signaling cues from dental mesenchymal cells facilitate further differentiation from presecretory to secretory ameloblasts. Secretory ameloblasts are polarized and secrete enamel matrix proteins, including amelogenin (Amelx) and ameloblastin (Ambn). Enamel crystal rods are formed and strengthened to mineralize the enamel matrix. After the enamel matrix is deposited, secretory ameloblasts differentiate into maturation ameloblasts. These cells are primarily responsible for ion transport and reabsorption of water and peptides hydrolyzed from the enamel matrix proteins to orchestrate the full mineralized enamel matrix. When enamel biomineralization is complete, ameloblasts subsequently apoptose. A single-cell transcriptome study indicates the two different types of ameloblasts that are distinguished by dentin sialophosphoprotein (*Dspp*) and *Ambn* [39]. The *Dspp*+ ameloblast modulates epithelial organization, whereas the *Ambn*+ ameloblast regulates enamel mineralization. Different TFs drive ameloblast differentiation at different stages, which we will describe in Section 3.3.

#### 2.2.5. Dental Epithelial Stem Cells (DESC)

Postnatally, adult SCs called dental epithelial SCs residing in the labial cervical loop (CL) regenerate dental epithelial cells for the continuously growing mouse incisors throughout the life of the mouse (Figure 1).

Dental epithelial SCs share several characteristics with other adult SCs in regenerative tissues such as discrete niche and the ability to differentiate [3,41]. Dental epithelial SCs are supported by a microenvironment in the CL (stem cell niche) that plays important roles in maintenance, proliferation, and cell fate decisions [42]. Dental epithelial SCs are identified by numerous SC markers, including *Sox2* [43,44], *Lrig1*, *Bmi1*, and *Gli1*. Dental epithelial SCs give rise to all the dental epithelial cells, including IEE, OEE, SR, and SI, during tooth development. IEE subsequently differentiate into ameloblasts, which secrete enamel matrix proteins (Figure 1). Transit-amplifying (TA) IEE cells are highly proliferative and migrate from the cervical loop toward the distal end of the mouse incisor. Recent combinatory analyses with single-cell transcriptome, in site hybridization and lineage tracing [36,39], revise previous concepts of dental SCs although a part of traditional classification for IEE, OEE, SR, SI is confirmed. For example, new study demonstrates that a highly proliferative population in IEE houses progenitor cells. However, they are different from previously reported stem cells residing in OEE, which are marked by *Sox2*, *Gli1*, *Bmi1*, *Lrig1* [36,39]. In this review, we will still use the traditional naming and markers but introduce recent modifications as appropriate.

## 3. The Role of TFs and Chromatin Regulators in Dental Epithelial Cell Fate

In this section, we describe various TFs and chromatin regulators that control dental epithelia at different stages of differentiation and different locations in the mouse mandible. We also provide the information about the mutations of these factors, which are associated with craniofacial diseases in humans, illustrating their clinical significance.

### 3.1. Epithelial Signal Centers at the Early Developmental Stage

During embryonic development, teeth are initiated from the dental lamina, a stripe of stratified epithelium first discovered at the sites of future tooth rows. Mouse embryonic dental lamina are characterized by localized expression of several TFs and signaling molecules, called epithelial signal centers. *Pitx2*, bicoid motif binding protein and a member of the paired-like homeobox family, arises in dental epithelium, and its expression persists in the developing tooth [45]. Pitx2 plays important roles in the pattern formation and differentiation of the tooth [46]. Mutations in the Pitx2 are associated with Axenfeld–Rieger Syndrome in humans, which presents with dental anomalies, including hypodontia and enamel hypoplasia [47]. *Sox2* marks a dental epithelial signaling center through interaction with *Pitx2 and Lef1* [48]. *Foxi3* [49], *Dlx2*, *Lef1*, and *p63* may also be responsible for driving dental fate [22]. *Foxi3* inhibits enamel knot formation [50] as its deletion leads to a supernumerary and incorrect pattern of cusps in the mouse [50]. The TF families of *Pax*, *Msx*, *Lhx*, and *Runx* are important during the early developmental stage as tooth development is arrested at the bud stage when *Pax9*, *Msx1*, or *Runx2* is deleted. The dental lamina stage is also disturbed when *Msx1/2*, *Dlx1/2*, and *Lhx6/7* are mutated [22]. Mutations of *PAX9* are associated with tooth agenesis in humans [51].

Nkx2-3, a member of the NK2 homeobox family of TF, also plays a critical role in the early developmental stage. Nkx2-3 mediates p21 expression and ectodysplasin-A signaling in the enamel knot for cusp formation during tooth development [52]. NK2 homeobox families are tissue-specific evolutionarily conserved TFs that regulate organ development, and Nkx2-3 has been identified as the dental epithelial specific Nkx factor through comparative microarray analyses. It may regulate dental SC fate because blocking *Nkx2-3* expands *Sox2* expressing populations in mouse organ culture system [52].

### 3.2. Adult SCs and Their Renewal

Sox2 has been recognized as a marker of dental epithelial SCs [36,43,53] and maintains competence for tooth formation [54]. Sox2 is critical for self-renewal of the SCs as conditional deletion of *Sox2* in the embryonic incisor epithelium leads to growth defects [54]. Tbx1 also controls proliferation and differentiation of dental SCs by modulating Pitx2 activation of p21. The deletion of *Tbx1* leads to loss of enamel formation in mice [55]. *Tbx1* is also a candidate gene for the 22q11.2 deletion syndrome causing dental defects in humans [56].

### 3.3. IEE/Ameloblast Lineage

*Sox2* and *Pitx2* controlling dental SCs also initiate the ameloblast lineage since conditional knockout (KO) mice for *Pitx2* [57] and *Sox2* [54] have defects in ameloblast development. Mutations of Pitx2 are identified in the Axenfeld–Rieger syndrome and tooth agenesis in humans [58].

AmeloD is a basic helix-loop-helix (bHLH) TF recently identified by screening a tooth germ complementary DNA (cDNA) library using a yeast two hybrid system [59]. The cell-type-specific class II bHLH TFs activate or repress gene transcription and control various organ morphogenesis, including muscle, neuron, and blood cells [60]. For example, muscle specific MyoD, a member of this class of TFs, has the capability to trans-differentiate fibroblasts to myoblasts [8]. The dental AmeloD regulates ameloblast differentiation from IEE and HERS cells [61]. AmeloD acts as a suppressor of E-cadherin and promotes the migration of dental epithelia. AmeloD is also important for the progression of the SCs towards enamel lineage since *AmeloD* KO results in enamel hypoplasia [61] and deletion of E-cadherin affects cell fate of SCs and progeny [62]. Mechanistically, AmeloD represses E-cadherin expression by transcriptional regulation, in which it directly binds to E-cadherin proximal promoter and recruits a chromatin repressive complex, including repressive histone H3K27me3 and Ezh2, which are a part of PRC2 core complex [59].

The chromatin organizer and TF, SATB1, controls ameloblast lineage at the early presecretory stage. SATB1 is a cell-type-specific gene regulator, originally found in T cells, in which it regulates gene transcription by folding chromatin into loop domains, and its deletion causes temporal and spatial mis-expression of numerous genes to arrest T-cell development [12]. In the tooth, SATB1 is expressed in presecretory ameloblasts and is essential to maintain ameloblast differentiation, cell polarity, and unidirectional secretion of matrix proteins [40]. *Satb1* null mice show thin and hypo-mineralized enamel, in which Amelx transports to the apical secretory front and secretion into the enamel space are impeded, resulting in a massive cytoplasmic accumulation of Amelx [40]. The expression of SATB1 is increased when secretion and processing of matrix protein are accelerated by overexpression of alternatively spliced Amelx, the leucine rich Amelx peptide [63].

Epiprofin (Epfn)/Sp6 is a key factor to promote IEE differentiation as well as proliferation [64,65]. Epfn/Sp6 is present in ameloblasts, including IEE and secretory and mature ameloblasts, with increasing levels of expression [61]. A missense variant in Epfn/Sp6 is associated with amelogenesis imperfecta in humans [66]. Ablation of *Epfn/Sp6* results in enamel defects during cusp and root formation in the mouse [65]. In contrast, over-expression of *Epfn/Sp6* in *Krt5*-expressing epithelia induces ectopic enamel in the lingual side of the incisor, where control mice do not normally form enamel [67]. Epfn/Sp6 controls enamel formation and tooth morphogenesis through the interaction of epithelial and mesenchyme [67]. Double KO mice for *Epfn/Sp6* and *AmeloD* show the transcriptional regulation by these two factors that is essential for epithelial cell invasion and cell proliferation [61].

At the maturation stage of the ameloblast, Runx2, runt-related TF2, is critical. Runx2 also regulates early tooth development [68]. Mutations in the *Runx2* gene cause dental defects [69] and the cleidocranial dysplasia syndrome in humans [70]. Conditional *Runx2* null mice show severe enamel hypo-mineralization [71].

### 3.4. Dental Lineage Plasticity

Dental epithelia are developmentally derived from ectoderm and separated from other ectoderm appendages such as skin epithelia. Dental and skin epithelia are distinct in their structure and function but share similar signal pathways and transcriptional machinery. However, the checkpoints to specify the dental lineage compared to epidermal ones are not well understood. Several studies show that the master regulators Sox21, Med1, and Msx2 govern the ectoderm lineages. Genetic removal of each factor re-programs enamel producing dental epithelia to epidermal/hair epithelia in transgenic mice, in which actual hair is generated in case of Sox21 and Med1 null incisors but not with Msx2 deletion.

Sox21 is a member of the SRY-Box (Sox) B group. Sox21 belongs to the SoxB2 protein family and functions as a transcriptional repressor although SoxB1 (Sox1–3) proteins are activators [72]. Sox21 was first found as a Sox2-associated factor [73]. The balance of transcriptional activation and repression is important for cell fates. For example, Sox21 repression of SoxB1 expression promotes neural differentiation [72]. Sox21 also regulates differentiation of hair cuticle, and *Sox21* null mice develop cyclic alopecia [74]. In teeth, Sox21 functions as a master TF to govern ectodermal lineages since conditional *Sox21* null mice switches the cell fate of dental epithelia to that generating hair, resulting in severe enamel hypoplasia [75]. *Sox21* null dental epithelial cells fail to commit to the ameloblast lineage. Instead, *Sox21* ablation leads to the formation of a unique microenvironment promoting hair fate because a part of dental epithelia are converted to mesenchymal like cells through epithelial mesenchymal transformation (EMT), which is supported by TGFβ [75]. These mesenchymal-like cells may generate a signal to stimulate epidermal differentiation as hair papilla do in the skin [61]. In addition, Sox21 ablation decreased E-cadherin expression, which is essential to maintain dental lineages [75].

Hair is also generated in the incisor of *Fam83h* null mice, although it forms relatively normal enamel [76]. The truncation mutations of FAM83H cause autosomal dominant hypocalcified amelogenesis imperfecta in humans [76]. The mechanism by which *Fam83h* ablation generates hair in the incisor may be related to ones for *Sox21* and *Med1*, although Fam83H is not a TF.

Msx2 also plays a critical role to control dental cell fate. Msx2 is a member of the family of divergent homeobox-containing genes. Msx2 was first reported as a transcriptional repressor [77]. It functions by forming heterodimers with other homeobox TFs such as CEBPα. Msx2 controls the cell fate of osteoblasts in bone and epithelium in ectodermal tissues such as skin, tooth, and mammary glands [78]. Msx2 antagonizes CEBPα and regulates ameloblast lineage by controlling expression of amelogenin [79]. Global *Msx2* KO mice do not form enamel in the normal location, and ectopic mineralization occurs in SR cells as a result of disturbing the differentiation of both ameloblast and SI at the maturation stage. Instead, *Msx2*-deficient OEE cells become highly proliferative and are transformed into epidermal cells. The epidermal and hair marker proteins are accumulate in the SR layer, but actual hair is not generated [80]. Therefore, Msx2 is considered a master TF, but its function may be dependent on the interaction with other unknown TFs.

Med1 also controls the cell fate of dental epithelia. Med1 ablation inhibits Notch1-mediated SI differentiation and disrupts amelogenesis essential for mineralization of the enamel matrix [20,21]. Med1 supports SI differentiation by directly facilitating Notch1-mediated gene transcription of Alpl by forming a complex with cleaved Notch1/Rbp-Jk on the *Alpl* promoter [20]. Instead, dental cells institute an epidermal program to regenerate ectopic hairs in the incisors. Sox2 expression persists beyond the CL and extends into the differentiation zone such that the cells within this zone remain multi-potent to maintain stem cell potentials [21]. These cells are induced to epidermal fate, likely by the calcium present in dental tissues [21].

The KO mice for these master regulators *Sox21*, *Msx2*, and *Med1* indicate the high lineage plasticity of dental epithelial cells. However, epidermal fate is derived through different types of enamel epithelia. Epidermal fate is induced through IEE/ameloblast, Notch1-expressing SI cells, and SR cells in KO mice for *Sox21*, *Med1*, and *Msx2*, respectively (Figure 2).

These results for *Sox21* and *Med1* null mice suggest both common and distinct mechanisms to underlying lineage plasticity. We propose that dental epithelial cells lacking these master regulators remain in an undifferentiated state and behave as pseudo stem cells in their location, where each factor is critical for their lineage. For example, *Sox21*-lacking and *Med1*-deficient dental epithelia fail to commit to their own fates for ameloblast and SI lineage, respectively. Instead, they may maintain multi-potency, as shown by the stem cell marker Sox2 extending into the differentiation zones in both *Med1* and *Sox21* null tooth [21,75]. These cells may be then re-programmed to skin epithelia through some stimulants present in their microenvironments. *Sox21* and *Med1* null tooth may utilize distinct stimulants since hair is generated in different locations of enamel organ. *Sox21* null mice generate hair in ameloblast zone [75], in which Sox21-deficient epithelial cells are converted to mesenchymal-like cells by EMT [75]. These mesenchymal cells may send a signal to induce epidermal fate as hair papilla do in the skin. In contrast, *Med1* null incisor generates hair under papillary layer, where the calcium is abundantly supplied from blood vessels. The extracellular calcium is transported for enamel mineralization there, but it may induce epidermal fate in case of *Med1* lacking tooth. Calcium induces epidermal fate of *Med1* lacking dental epithelial cell in culture [21], and calcium gradient stimulates epidermal differentiation in the skin [81].

This re-programming may be driven by chromatin dynamics. Cell fate is supported by super-enhancers in ESC and somatic cells [82,83], in which the Mediator complex and the fate TFs are also densely incorporated [84]. Our recent results show that the same is true in dental epithelial SCs. Med1 may regulate cell fate by forming the super-enhancers, in which dental enamel fate TFs are highly incorporated (unpublished observations). Med1 ablation blocks the enamel lineage, resulting in enamel hypoplasia [21] (Figure 3A microCT panels [21] by disturbing these epigenetic regulation (unpublished observations). We present a model to show that dental cell fate is controlled by epigenetic processes: (1) Mediator complex containing Med1 (blue) forms super-enhancers, (2) several fate TFs such as Pitx2 or Sox21 (pink and green) are densely recruited into the super-enhancers, (3) the super-enhancers are linked to the promoter of dental specific genes as some of Mediator subunits bind to general transcriptional complex (yellow), and (4) gene transcription for enamel lineage is induced through RNA polymerase (PIC) (Figure 3B). Therefore, fate TFs such as Sox21 or chromatin regulator of Med1 may be essential for enamel formation by controlling dental epithelial cell fate.

Fate TFs are present in specific locations of dental epithelia where they function. However, Mediator complexes are ubiquitously expressed in all cells as universal transcriptional machinery to support the function of these fate TFs. In fact, Med1 deletion from *Krt14*-expressing epithelia converts the cell fate not only in dental epithelia but also in skin, where it controls the balance of three epidermal cell fates involving hair, sebaceous gland, and interfollicular epidermis [85]. Sox21 is not a universal TF but is present in both dental and skin epithelia, and *Sox21* deletion from *Krt14* epithelia also affect cell fates of hair keratinocytes in the skin [74].

Functions of various TFs and chromatin regulators in enamel organ are summarized in Table 1. The function is deduced through mouse phenotypes of either global or conditional KO mouse models or enamel organ culture lacking the factors. The localization within the enamel epithelia is shown. Their potential roles in DESCs are also proposed through histological or gene expression analyses of mouse or organ culture models by focusing stem cell functions such as maintenance/proliferation and lineage commitments. The cell fate decision means DESCs commit to dental epithelial fate to produce the enamel but suppress non-dental ectoderm lineages towards hair and epidermis.

## 4. Conclusions

In summary, we have discussed recent progress in our understanding of the TFs that control dental epithelial cell fate. We also demonstrate the high plasticity of dental epithelial cells that can be re-programmed to other lineages by manipulation of the expression of master regulators. These factors are obviously candidates for cell re-programming, in which one factor or combination of factors is capable of converting either induced pluripotent stem cell (iPS) or other somatic cells into dental epithelia to produce enamel. Further investigation of their mechanisms at the genetic level will advance our efforts to generate new teeth.

## Figures and Tables

**Figure 1 ijms-21-08952-f001:**
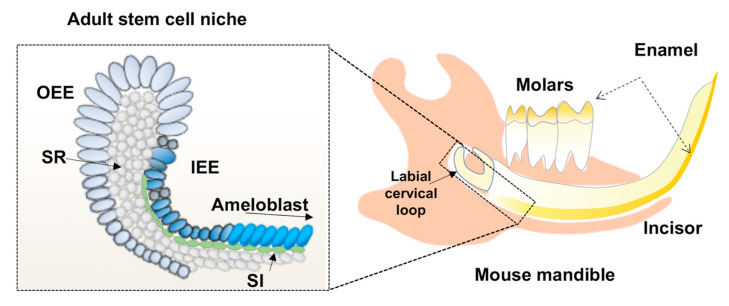
A diagram to show the location of adult stem cell niche in mouse mandible, where stem cells (SC) and different type of dental epithelia reside such as outer and inner enamel epithelia (OEE, IEE), the stellate reticulum (SR), and the stratum intermedium (SI). The IEE are differentiated into ameloblast through sequential steps and generate enamel on the tooth.

**Figure 2 ijms-21-08952-f002:**
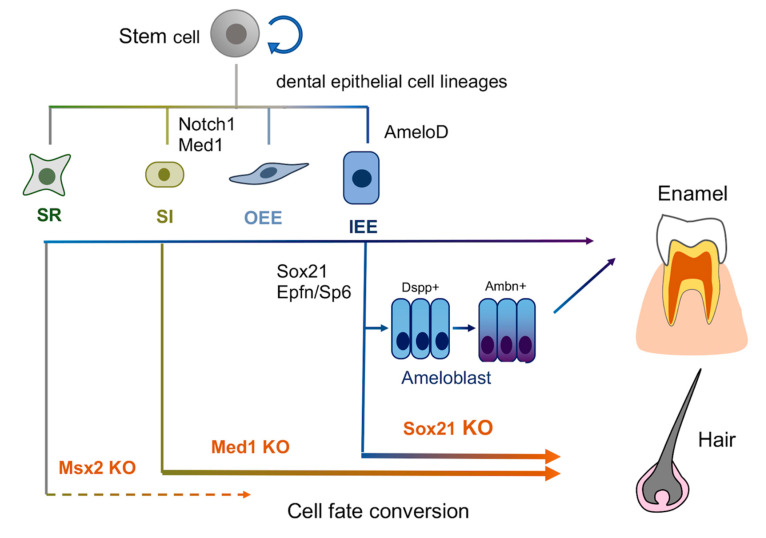
A proposed model in which Sox21, Med1 and Msx2 knockout (KO) mice alters dental epithelial cell fate to skin epithelia through IEE/ameloblast, SI and SR lineage, respectively.

**Figure 3 ijms-21-08952-f003:**
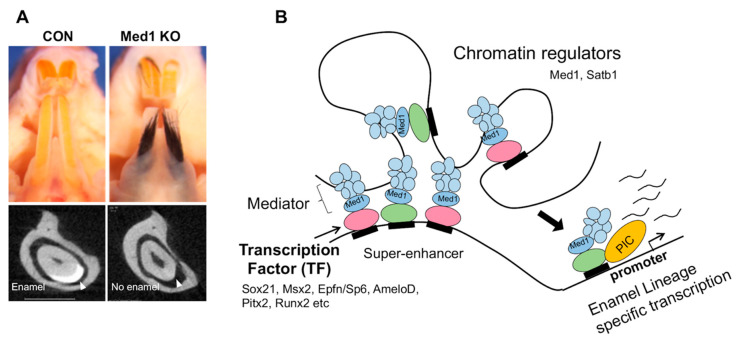
(**A**) Med1 KO mice generate hairs in incisors while disrupting enamel formation. (**B**) A proposed model to illustrate a role of transcription factors and chromatin regulators to control enamel lineage specific transcription.

**Table 1 ijms-21-08952-t001:** The function of transcription factors and chromatin regulators in enamel organ and their potential role in dental stem cells. The localization of these factors is also shown.

Regulator	Function in Enamel Organ	Localization	Potential Role in DESC	Reference
Transcription factor				
AmeloD	enamel formation	IEE	ameloblast differentiation	[59,61]
Epfn/Sp6	enamel formation	IEE, ameloblast	IEE differentiation/proliferation	[64,65]
Foxi3	molar crown patterning	DE, DESC	epithelial differentiation	[49]
Lef1	early development	DE	dental epithelial cell fate	[22,48]
Msx2	ectodermal development	DE, ameloblast	differentiation of ameloblast and SI	[78,80]
Nkx2-3	cusp formation/early development	DE	p21 expression and EDA signaling	[52]
Pitx2	pattern formation/differentiation	DE	ameloblast lineage	[45,46]
Runx2	early development	DE, ameloblast	ameloblast differentiation	[69,71]
Sox2	stem cell maintenance	DESC	maintainance of stemness	[53]
Sox21	enamel mineralization	IEE, ameloblast	dental epithelial cell fate decision	[75]
Tbx1	early development	DE	proliferation and differentiation	[55]
Chromatin regulator				
Med1	enamel mineralization	DESC, SI	dental epithelial cell fate decision	[20,21]
Satb1	enamel mineralization	preameloblast	ameloblast lineage	[40,63]

DESC dental epithelial stem cells, DE dental epithelial signaling center in early development, IEE: inner enamel epithelia, SI statum intermedium.

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
