# Peer review of "Transcriptional Regulation of Dental Epithelial Cell Fate"

_ijms, 2020, doi:10.3390/ijms21238952_

Round 1

Reviewer 1 Report

Abstract:

- "Tooth enamel is regenerated by dental epithelial stem cells and progeny residing in the tooth" - this is valid only for some groups of animals, e.g. rodents, not for humans

Introduction:

  • Lines 32-40: no reference is given here
  • Lines 43-48: TFs have a plethora of roles in every biological process as they are the ultimate regulators of gene expression; the authors should revert the orders of the topics, i.e. not introduce TF importance because they can reprogram somatic cells

2. Development and morphogenesis of mouse dental epithelium

  • Conceptually, the authors should first discuss the paragraphs 2.2.2, 2.2.3, 2.2.4, and only afterwards the adult dental epithelial stem cells - events described 2.2.2-2.2.4 happen during development, and they are then maintained by DESCS postnatally in the incisor

3. The role of TFs and chromatin regulators in dental epithelial cell fate

  • Is Fam83H a transcription factor?
  • "These results suggest both common and distinct mechanisms to underlying lineage plasticity. We propose that dental epithelial cells lacking these master regulators remain in an undifferentiated state and behave as pseudo ectodermal cells in their location,where each factor is critical for their lineage. These cells are then re-programmed to skin epithelia through some stimulant present in their microenvironment" - as written now, the review does not clearly support this model.
  • "Here, we present a model to show that dental cell fate is determined by chromatin dynamics in which several fate TFs such as Pitx2, Satb1, Tbx1, Runx2, Sp6, are densely recruited with Mediator into super-enhancers, facilitating dental specific gene transcription (Figure 3).Therefore, we may be able to control dental fate by manipulating these factors and dental specific chromatin dynamics"  - similar as above: the logical steps from the evidence exposed to the model proposed are not clear.
  • More schematic figures should be provided, to clarify the different roles of the TFs discussed on DESCs fate determination

Author Response

We answer for reviewer’s comments as follows. We used the line numbers in final version of revised manuscript, that is different from ones in the file showing tracking changes by color.

Reviewer 1 Comments and Suggestions for Authors

Abstract:"Tooth enamel is regenerated by dental epithelial stem cells and progeny residing in the tooth" - this is valid only for some groups of animals, e.g. rodents, not for humans

Answer: We agree. We removed the word of the “stem cells” from this sentence (line 15). Instead, we describe the stem cells and progeny in other sentences to describe the mouse models (line 19-20).

Revised line 15: Dental enamel is hardest tissue in the body and produced by dental epithelial cells residing in the tooth.

Revised line 19-20 In this review, we describe current understanding of these regulators essential for regeneration of dental epithelial stem cells and progeny, that are identified through transgenic mouse models.

Introduction:

Lines 32-40: no reference is given here

Answer: We added the references 1-3 in line 32-37.

Lines 43-48: TFs have a plethora of roles in every biological process as they are the ultimate regulators of gene expression; the authors should revert the orders of the topics, i.e. not introduce TF importance because they can reprogram somatic cells

Answer: We changed the order of paragraphs. We added new sentences to introduce the general function of TF in line 38-42. We moved the sentences to describe the rational of the studies “Understanding…” to the last paragraph of the introduction (line 70-73).

  1. Development and morphogenesis of mouse dental epithelium

Conceptually, the authors should first discuss the paragraphs 2.2.2, 2.2.3, 2.2.4, and only afterwards the adult dental epithelial stem cells - events described 2.2.2-2.2.4 happen during development, and they are then maintained by DESCS postnatally in the incisor

Answer: We agree and changed the order to follow developmental processes and postnatal regeneration through stem cells. In addition, we also revised the introduction to show these changes (line 76-79).

  1. The role of TFs and chromatin regulators in dental epithelial cell fate

Is Fam83H a transcription factor?

Answer: No, Fam83H is not. We still keep the sentences but added the explanation that the similar hair phenotype is observed in Fam83 KO tooth, in which the mechanism may relate to ones for Sox21 or Med1.

Revised: The mechanism by which Fam83h ablation generates hair in the incisor [75] may be related to ones for Sox21 or Med1, although Fam83H is not  a TF. (line 262-263)

"These results suggest both common and distinct mechanisms to underlying lineage plasticity. We propose that dental epithelial cells lacking these master regulators remain in an undifferentiated state and behave as pseudo ectodermal cells in their location,where each factor is critical for their lineage. These cells are then re-programmed to skin epithelia through some stimulant present in their microenvironment" - as written now, the review does not clearly support this model.

Answer: We support the model by specifically comparing two mouse models for Sox21 and Med1 KO. We added the detailed explanations (line 289-) for the common mechanism for Med1 and Sox21, in which undifferentiated cells behave as stem cells to maintain multi-potency. We also describe the distinct stimulants to induce epidermal fate, that is present in the microenvironments for Med1 and Sox21 KO tooth (line 296-). We propose the different stimulants as hair generates in different locations of the enamel organ in two models. We propose the calcium for Med1 KO and EMT mediated mesenchymal cells for Sox21 KO. We added the references to support the model (line 289-304).

"Here, we present a model to show that dental cell fate is determined by chromatin dynamics in which several fate TFs such as Pitx2, Satb1, Tbx1, Runx2, Sp6, are densely recruited with Mediator into super-enhancers, facilitating dental specific gene transcription (Figure 3).Therefore, we may be able to control dental fate by manipulating these factors and dental specific chromatin dynamics"  - similar as above: the logical steps from the evidence exposed to the model proposed are not clear.

Answer: We explained our hypothetical model by following logical steps. First, we describe the universal mechanism, in which cell fate is controlled by super-enhancers and fate TFs that are densely recruited with Mediator, which is first demonstrated in ESC and some somatic cells as noted in the references listed  (line 305-). Secondly, we show that the same is true in dental epithelial SCs by introducing our unpublished data (line 307-).

We then described four different epigenetic processes to lead enamel lineage specific transcription by adding detailed explanations of Figure 3B (line 400-406).

We moved the sentences to mention the universal function of Sox21 and Med1 in both tooth and skin after explaining Figure 3 (line 318-).

More schematic figures should be provided, to clarify the different roles of the TFs discussed on DESCs fate determination

Answer: We added a new Table to show the different roles of the TFs in DESC as well as dental epithelial cells such as ameloblasts (line 333-334). Reviewer 2 requested the Table to show the different roles of the TFs, that would provide similar information compared to the figure requested by reviewer 1. Therefore, we present the Table instead of schematic figure. If the reviewer still prefers the figure, we are able to provide it.

Reviewer 2

Comments and Suggestions for Authors

In this mini review Yoshizaki et al. critically evaluated the role of fate determining transcription factors and chromatin regulators for regeneration of dental epithelial stem cells and progeny. The authors covered all topics comprehensively starting from development and morphogenesis of mouse dental epithelium to the function of critical factors in stem cells or progeny to drive enamel lineages.

I recommend this review for publication with only one suggestion/addition.

The author should add a table which will show all the TFs involved in regeneration of dental epithelial stem cells (DESCs) with their possible regulatory role. This will provide an ease for the readers/researchers in getting the updated information just looking to the table.

Answer: We added the Table as indicated (line 333-334). We also added the description of the Table (line 327-333).

For example

S.No

Transcription factor

Role/function

Possible role in DESCs

Reference

1

Pitx2

pattern formation and differentiation of the tooth

Lin et al., 1999

Reviewer 2 Report

In this mini review Yoshizaki et al. critically evaluated the role of fate determining transcription factors and chromatin regulators for regeneration of dental epithelial stem cells and progeny. The authors covered all topics comprehensively starting from development and morphogenesis of mouse dental epithelium to the function of critical factors in stem cells or progeny to drive enamel lineages.

I recommend this review for publication with only one suggestion/addition.

The author should add a table which will show all the TFs involved in regeneration of dental epithelial stem cells (DESCs) with their possible regulatory role. This will provide an ease for the readers/researchers in getting the updated information just looking to the table.

For example

S.No

Transcription factor

Role/function

Possible role in DESCs

Reference

1

Pitx2

pattern formation and differentiation of the tooth

Lin et al., 1999

Author Response

Reviewer 2

Comments and Suggestions for Authors

In this mini review Yoshizaki et al. critically evaluated the role of fate determining transcription factors and chromatin regulators for regeneration of dental epithelial stem cells and progeny. The authors covered all topics comprehensively starting from development and morphogenesis of mouse dental epithelium to the function of critical factors in stem cells or progeny to drive enamel lineages.

I recommend this review for publication with only one suggestion/addition.

The author should add a table which will show all the TFs involved in regeneration of dental epithelial stem cells (DESCs) with their possible regulatory role. This will provide an ease for the readers/researchers in getting the updated information just looking to the table.

Answer: We added the Table as indicated (line 333-334). We also added the description of the Table (line 327-333).

For example

S.No

Transcription factor

Role/function

Possible role in DESCs

Reference

1

Pitx2

pattern formation and differentiation of the tooth

Lin et al., 1999

Round 2

Reviewer 1 Report

The authors replied properly to the comments.